# Reaction Time Improvements by Neural Bistability

**DOI:** 10.3390/bs9030028

**Published:** 2019-03-18

**Authors:** Henk Koppelaar, Parastou Kordestani Moghadam, Kamruzzaman Khan, Sareh Kouhkani, Gijs Segers, Martin van Warmerdam

**Affiliations:** 1Faculty of Electrical Engineering, Mathematics and Computer Science, University of Delft, 2628 XE Delft, The Netherlands; 2Social Determinants of Health Research Center, Lorestan University of Medical Sciences, Korramabad 6813833946, Iran; Kordestani.p@lums.ac.ir; 3Department of Mathematics, Pabna University of Science and Technology, Pabna 6600, Bangladesh; k.khanru@gmail.com; 4Department of Mathematics, Islamic Azad University Branch of Shabestar, Shabestar 5381637181, Iran; skouhkani@yahoo.com; 5Sports Trainer, Physical Education Teacher, Visual Performance Trainer, Gymi Sports, 4907 BC Oosterhout, The Netherlands; gijs.segers@ziggo.nl; 6Van Warmerdam Optometry’s, 5234 KA Hertogenbosch, The Netherlands; Martin@vanwarmerdam.nl

**Keywords:** eye-hand coordination, FitzHugh-Nagumo model, sensorimotor system, reaction time, stroboscopic training

## Abstract

The often reported reduction of Reaction Time (RT) by Vision Training) is successfully replicated by 81 athletes across sports. This enabled us to achieve a mean reduction of RTs for athletes eye-hand coordination of more than 10%, with high statistical significance. We explain how such an observed effect of Sensorimotor systems’ plasticity causing reduced RT can last in practice for multiple days and even weeks in subjects, via a proof of principle. Its mathematical neural model can be forced outside a previous stable (but long) RT into a state leading to reduced eye-hand coordination RT, which is, again, in a stable neural state.

## 1. Introduction

Exercise physiology and psychology, biochemistry, and biomechanics in sports have caused long lasting performance improvements in sports. However, the science of motor control in sport practices has not come to full maturity until today even though it was proposed in the IOC Congress on Sport Sciences [1] as early as 1996. This paper explains sensorimotor Reaction Time (RT) improvements of more than 10% of fully trained athletes even at the top of their abilities. The explanation purports the ability of the human neural system to become stabilized in more efficient control of the motor system, simply by vision training.

The societal impact of this research is related by Burris et al. [2], by way of Ted Williams’ observation “I think, without question, the hardest single thing to do in sports is to hit a baseball.” Burris and colleagues [2] explain “Major League Baseball (MLB) pitches move at speeds near the processing limits of the vestibular-ocular tracking, which leaves the batter with mere milliseconds to decipher the pitch, project its trajectory, decide to swing, and coordinate the timing and trajectory of a 2.25-inch diameter bat. The immense difficulty of this task is underscored by the fact that players who hit successfully on less than a third of their at-bats can receive one-hundred million dollar contracts in today’s free-agent market.”

Room for improvement of RTs is from Rathelot et al. [3]. They proved that neural fields competing in visual perception versus dexterous command may converge into a single command mode, which will induce extremely short reaction times, which are shorter than commonly achieved. Their finding is clearly task dependent [4,5,6,7] and sustained by Weiler et al. [8]. We exploit by proof of principle in Section 2 and Section 4 (via a mathematical FitzHugh-Nagumo model of the sensorimotor system, for short FHN) that a reduction of RTs is stable in a new states of mind outside their previously obtained RT, which was a comfort behavioral zone of athletes. The model explains how improved eye-hand reaction times after a tailored “More Input Eye vision is the Key” (MIEK) training occur using LCD shutter glasses, see Section 3. The reduced RTs observed after the training are of high statistical significance relative to the RTs before the training, as reported in Section 3 and Section 4.4.

Athletes, like all humans, miss over 9% of their visual inputs because of spontaneous eyeblinks. In the tailored MIEK Visual Performance training, we enforce eyeblinks up to much higher rates using LCD shutter glasses (operating similar to eye-blinks and called ‘strobospectacles’). The strobospectacles train athletes’ eyes to intermittently enforce taking a break, reducing visual input, and, thereby, decreasing redundant feedback [9]. The reduced RT by strobotraining has been found to last beyond the training period but is, currently, largely unexplained. This novel explanation is the goal of this paper. We concentrate on the motor control by neural fields. The sensor part of the human system deserves further scrutiny, because speeding up RTs has also been found by invisible stimuli.

Strobotraining naturally speeds-up RTs by involving the phenomenon of invisible stimuli [10] and by establishing stable memory effects [9,11,12,13,14,15,16]. We conjecture that strobotraining as a side effect invokes the human ability speeding up RTs by invisible stimuli, because of the fast shutting-off light. For example, experienced ball athletes do not follow the ball’s trajectory, but rather observe the initial movement of the ball, remember it, and, with a rapid eye saccade, go to a projected final position of the ball’s path. Memory effects are perceivable via EEG patterns [17,18], which are emanating from oscillating neural fields. Figure 1 depicts how to move a sensorimotor FHN 1 ability to a new field of improved ability FHN 2. The neural oscillations were correctly hypothesized by some authors as memory effects [9,13,14,16]. For more extensive information about FHN oscillations, see References [19].

By hypothesizing an idea complimentary to Schmidt’s team [20] for cognitive learning, we show that strobotraining results in a speed of motor response moves from a habitual zone in athletes’ sensorimotor system, i.e., by physical training achieved shortest RT, to a new stable neural state with about 10% faster RT than the old achieved habitual one. Note that the issue here is: without physical muscle training, a sensorimotor system change occurs. This is comprehensively reviewed by Ahissar [21]: perceptual training is aimed at modifying sensorimotor abilities. His report stems from auditorily perceptional prior work [22], followed up by Ho and coworkers [23], who also show oscillating neural correlates, which are different for left and right sides of the body. Later research applies the modelling to developing brains of adolescents [24].

The main proof by principle in this paper, by using a mathematical FitzHugh-Nagumo model, is that such old and new stabilities, in short ‘bistabilities,’ occur through two interacting models: from a habitual zone, via a meta-stable intermediate state, to a new stable zone. This type of modelling for research is common in physics and is reviewed for broader application by Kriegeskorte et al. [25].

The findings enforce Güllich’s research among top talents in sports [26] who revealed that world champions do train in many disciplines. As such, they learn more than one neural comfort zone.

## 2. Sensorimotor Tasks Induce Reaction Delays

Rathelot, Dum, and Strick [3] discovered that the posterior parietal cortex contains a common command apparatus for hand movements. If employed, this command reduces RTs to an optimum. Of course, the central nervous system has many sources of command to the motor apparatus of the human body. Such plasticity [3] of the sensorimotor system has been studied for the aural system [21,22] and has a central role for the visual system. The brains’ plasticity is extreme, even noise in neurons is sensible to signals [27]. Because the many sources of command [28,29] researchers advocate a dual task paradigm: two separate neural fields with one for the oculomotor system and one for the motor behavior of the limbs. Haak et al. [30] defend three systems, which are simulated by Ambrosio et al. [31]. Three systems may occur if proprioscepsis is considered. For instance, Ebsch and Rosenbaum [32] show that cortical circuits can respond to natural and artificial stimuli in the context of locally gained imbalance, which is this local behavior (i.e., the neural field) that is relocatable in the brain and, hence, can be untrained.

To make our analysis more tractable, we simplify the neural fields in the sensorimotor system, according to Graham [12,33], which is commonly done in neural modelling. We split it in an oculomotor system and a motor part, such that, if the first is sufficiently stimulated, then the subsequent motor response starts [32]. These two systems can alter their modus operandi to operate independently, or dependently but separately [6], or even under a common command [3]. Examples of various stabilities are: hand or foot movement does not change much if subjects either see or do not see their hand or foot, so this suggests independence of the two effector systems. Reaching for a glass of water requires executing hand-eye movements in fine temporal coupling between the two effectors. In sum, a large repertoire of eye-hand movements in our daily lives might use such different ‘circuitries’, namely neural fields. For instance, in smooth pursuit tasks, such as throttling, pilots in the mean have slightly shorter delays [34] than ball sports athletes, which corresponds to Reference [3].

Plasticity brings us from one state to the other. Jana et al. [4,6] hypothesized an executive controller assessing behavioral context, which allows switching between modes or merging of modes. To paraphrase Schmidt et al. [35] in response to a transient input from strobospectacles, the sensorimotor system relaxes to a single new state. Cooperation between different neural fields is a necessary reduction of computational complexity for large networks, according to Kaminski [36] and Stefanescu and Jirsa [37]. From our model (see Appendix A, Equation (A1)), propagating waves in detail are elicitable. Figure 1 shows a gradually spreading traveling wave [38] from FHN 1 reaching another field FHN 2. This enables mapping cortical wave patterns to the original point of FHN spikes trains. Such research has been conducted for smooth pursuit tasks [39].

Wave speed such as propagation between neural fields has been studied by Hafed et al. [40]. The strobotraining experiments, carried out in our analysis (Section 3), enforces athletes an uncommon way of processing visual input, i.e., enforced blinking. Previous research has proved the neural impact of blinking [41,42,43]. Thereafter, stable oscillations of the newly achieved FHN 2 state exist without continued input.

The principle we seek out to prove is that: speed-up of the sensorimotor system may happen because at least two stable states for the two effectors exist including each with different time delays, as depicted in Figure 2. The time delay of the coupling *C* between the two groups of sensorimotor neurons FHN1 and FHN2 is $\tau_{C}$. Internal feedback *K* within the sensorimotor neurons group is $\tau_{K}$. The simplified model neglects the internal feedback *K* to an apparatus of the body (enclosed by the dotted line) because of the complexity of the visual effector. Why to exclude? Because it includes feedback from one group of sensorimotor neurons to the eyes. For example, feedback to the eyes to express saccades after the hand moves a ball [44].

## 3. Materials and Methods

Basic comparison material was established using measurements from training elsewhere in the world. All publications on strobotraining were collected. If such training is done adequately, assessments [2,46] show that improvements may occur in nine tested abilities of students and stamina of the visual system. Positive results also appeared for toss-to-wall [46], baseball [2,13,47,48], basketball [49,50], cricket, football [9,51,52], fencing, frisbee [9], hockey [15,53], rugby [54], softball [55,56], tennis, table tennis [57], volleyball [49], and water-polo [49]. Outside of sports, improvements are known for motion sickness [58], older adults [59], and visually impaired youth [60]. The opposite of enforcing eye-blinks, i.e., stroboscopic lightning, prevented visual sharpening in fish [61]. A full overview including RT measurement equipment up to 2007, is from Erickson [62] and is extended later by Ellison [53] in his dissertation. There are differences in results between measurement techniques via virtual or real ball grasping [63].

To prove stability of shorter RTs, we use a bistable Fitzhugh-Nagumo model for the eye-hand coordination. The main idea of this explanation is the plasticity of the neural system, whose properties enable the un-training of old and learning novel behaviors. History of sports shows the relentless shift to shorter RTs. Otherwise, records would not be broken. Most of RT improvements comes from eye-hand coordination, i.e., the sensorimotor system. The RT improvement is achieved by shifting the control of the motor system to a new -albeit stable- neural field (otherwise, the RT would not be shorter and the new and faster behavior could not be repeated). The ultimate speed is achieved if the shift of the neural control is made into the cerebellum [3]. The learning example is a novel behavior that starts in a neural field in the right hemisphere. After training, it moves to the left hemisphere and, after full mastery, it resides in the cerebellum. Un-training a habit or behavior neurologically is mostly the same as training, i.e., shifting a neural field. An example of un-training (in perception) is: ignoring task irrelevant stimuli [64]; or, un-training the checkup of ambiguous features [65]. Both examples are replicated by our MIEK Visual Performance Training and reduce RTs of the sensorimotor system as is displayed clearly by the green versus red lines in Figures 4–6 below.

### 3.1. Participants of the Experiment

The current study was conducted with 81 athletes, men and women aged 19–40, from eight sport areas: the (Dutch) National Hockey League, Tennis, Squash, Taekwondo, Fencing, Snowboarding, and the top Umpires of two hockey leagues (KNHB and FIH). Testing and training were completed at the teams’ competition sites between 2014 and 2019. One of the authors (GS) and the team’s visual performance trainer conducted all trainings. Participation was voluntary, and no benefits or penalties were offered to encourage participation. The potential effects of the training were explained to participants. All participants had informed consent to the training.

### 3.2. Measurements

All athletes participating in the experiment underwent baseline measurements. Baseline measurements were conducted by authors MW and by GS. Followed MIEK Visual Performance Training once per week over the course of 12 weeks, which was conducted by GS. Each training totaled about 90 min, of which 30 min was completed with the stroboglasses. Lastly, the athletes underwent posterior measurements to determine progress or deterioration relative to the baseline measurement, which was also conducted by MW and by GS.

Ellison [53] checked embeddability of testing methods for various sports (see Reference [62] with an update in 2016 [66]). Based on their assessment check, we did not find any evidence that would bias our testing method.

### 3.3. Computational Equipment

The FitzhughNagumo model was run on a Quadcore ASUS laptop, using Maple software, version 18 from MapleSoft, Waterloo, ON, Canada.

## 4. Results

### 4.1. Models with at Least Two Stable States Depict Learning

Graham [33] reviews the success of the simple decision rule model, which accounted for so many experimental results and represented most areas of visual cortex and the rest of the brain. In response to near threshold patterns, only a small proportion of neural neurons (‘analyzers’ in Reference [33]) are being stimulated above their threshold, as built in Equations (1). Retention or memory of an improved reaction time (RT) in this mathematical model is proved by its second stable state, which is different from its first stable state, i.e., its ‘habitual zone’. It started in 1961 (FitzHugh [67]) and in 1962. When Nagumo et al. [68] proposed a model for emulating the current signal observed in a living organism’s excitable cells. This is coined by the name ‘FitzHugh-Nagumo’ (FHN) model of mathematical neuroscience and is a simpler version of the Hodgkin-Huxley (HH) model of spiking neurons. Graham’s [33] hypothesized decision variable (DV) models the decision to move after the signal from a sensor system. The DV is an abstract construct that has very successfully predicted RTs distribution [69]. An independent source of insight for this is are experiments to study neural adaptation in conceptual processing [70]. Downstream activation from the perception to action in cognition even shows the RT assessment and even the decline of training effects we discuss here. Therefore, the DV mediates between eyes and hand, i.e., two mean fields of neurons. This leads, in our model, to a bifurcation of states or, said otherwise, the training brings the model to a critical value at which a system is unstable and, by further training, state variables bifurcate become stable again. The DV then starts the motor system to move earlier than before, in a previous state. Subsequent training then stabilizes this newly attained achievement as depicted by green lines in Figures 4–6 below.

### 4.2. Complimentary Work

Bistability is a well-known phenomenon such as the sleep-wake rhythm. This is possible because the brain has both stable states, but we cannot be in both states simultaneously. Therefore, it is bi-stable. Bistable perception is also well-known [71,72,73,74,75]. Exogenous measures such as disturbing sleep or control can bring or force a neural system from one to another stable state and back for some period [20,72,74,75]. Input from perception gradually build-up transitions of motor populations, until the competition between alternative representations is resolved by a neural threshold mechanism. Bistable neurons reflect neural collectives rather than properties of individual populations or neurons. Such bistability may occur even at a single neuron level [76] and at the level of lattices and fields. The single neuron approach to modelling complies with multiple neuron models, as explained by Graham [12,33]. We complement the pioneering work done by Schmidt et al. [20] to explain learning cognitive tasks by employing the plasticity of the brain to model shifting neural fields. A curiosity of difference between Schmidt et al. [20] and our work is the difference between the employed mathematical models. Prior work by Montbrio and Roxin [77] developed models to bridge the gap between large networks of spiking neurons and their observed firing rate. They need observations from EEGs to gauge cognitive learning tasks, which we do not pursue. They introduced a second order differential equation. With this, we depart from the FHN model [78], as inspired by Hodgkin and Huxley [79] with a higher order nonlinearity. The difference between the approaches is reminiscent of statistical physics and the ensemble theory for many neurons by Schmidt et al. [20] versus our FHN full causal differential equations (see also Rankin and Chavane) [80]. Our choice is supported by Chen and Majda [81] who invoke dimension reduction techniques for FHN ensemble models.

### 4.3. The FitzHugh-Nagumo (FHN) Model

We consider the following system [82,83] of FHN equations.
(1)$$\begin{array}{l} \varepsilon \frac{\partial x_{1}(t)}{\partial t}=x_{1}(t)-\frac{1}{3}x_{1}(t{)}^{3}-y_{1}(t)+C[x_{2}(t-\tau )-x_{1}(t)]\\ \frac{\partial y_{1}}{\partial t}=x_{1}(t)+a,\\ \varepsilon \frac{\partial x_{2}(t)}{\partial t}=x_{2}(t)-\frac{1}{3}x_{2}(t{)}^{3}-y_{2}(t)+C[x_{1}(t-\tau )-x_{2}(t)]\\ \frac{\partial y_{2}}{\partial t}=x_{2}(t)+a. \end{array}$$

The neural system with subscript 1 models, the visual part of the sensorimotor system, and subscript 2 models the motor part. The RT τ emerges because of a delay between the two neurons. Observations over 81 athletes (see below Figures 4–6 of the experimental results) showed that the RT is reduced by 10.8%. This means it is shifted because of the training in the mean from 0.69 → 0.61 s of RTs for the athletes’ sensorimotor system.

Equation (1) has three stable points, as seen from left to right in the phase plot of Figure 3, at the zero crossings of the curved line. The system is started from a stable state at the origin, moves to the first unstable state at the first crossing to the right, and is pushed further by excitation to the second stable state at the point *x* = 1.8, which is again a stable state.

The two-effector model can evoke a speed-up in RT if it bifurcates to a novel stable state. Evidence for such a scenario is independently gained from various sources [37,84,85] by studying neural fields. We take two neurons 1 and 2 along with the generic two-variable reaction–diffusion model for excitable and inhibitory media—the FritzHugh-Nagumo model (1)—such that the excitory field 1 is modeled with a membrane potential $x_{1}$ at time n. The electric membrane property is captured by ε and $x_{1}(0)$, which is the potential of the membrane at rest at the starting time zero of excitation (see Kaminski et al. [36]).

Memory effects of such novel obtained bi-stable states last for days and weeks (variable among subjects) and are characterized by time-scales of magnitude larger than those of perceptual reversals [72,74]. We study sensorimotor dynamics through the lens of a mathematical model. The new stable state is shown in Figure 2. An analog electrical model simulating the FHN equations gave the same stability result [86], based upon FitzHugh’s prior model [87].

The delay time τ is in the brain, which is not necessarily symmetrical (forward versus feedback). We make it symmetrical [78] because we only need proof of two different feedforward delays including one measurement prior to and one after the MIEK Visual Performance Training. Moreover, Panchuk et al. [88] give proof to convert an FHN system with asymmetric delays into a system with one delay.

This plot of Figure 3 in the phase plane shows a nullcline of a single FHN system passing three stability points at the horizontal axis including the zero points of the curve, which are the stable states with the middle one in an intermediate meta-stable state. The phase lines of the system are omitted. In a full phase plot of hundreds of behaviors, the stable points would be difficult to detect. In spite of the variability of neurons’ spiking behavior, Panchuk et al. [45] discover an invariant to detect coherent spiking of the two FHN systems if the number of involved neurons is small. Otherwise, coherent solutions cannot be realized due to the refractory phases of the subsystems.

### 4.4. Results of the Experiments

The results of the experiments show in Figure 4, Figure 5 and Figure 6.

## 5. Conclusions

Bi-stability is shown to be feasible in the applied FHN model, which also shows the retention of the obtained changes.

The exact shape of fields of cooperating neurons is variable and not fully classified. Researchers [89,90,91] defend ring-like architectures of cooperating FHN neurons. Neural field propagation happens in various ways, such as a non-ring traveling wave (see Figure 1). We do not follow [90,91] because they do not include a propagation delay.

The explanation of reduced RTs by strobotraining might need a complimentary explanation. As is well known, dexterous behavior depends upon the context of the task at hand. It might well be that the stroboscopic training forces athletes out of their habitual or comfort zone by perceived similarity or reminiscence of the training with known and familiar tasks. This phenomenon is discussed by Güllich [26]. He discovered that training of only one stable state (comfort zone) is less productive to achieve the world top than to train many sports, or in our model terms, to train many stable states.

What still has to be done is understanding the context of multiple tasks (sports) and its inputs. The quenching of neural variability was discovered by Churchland [92] and explained by Chang [93] in terms of reverse engineering, and the optimal dose of input to maximize the effect/output. Rathelot et al. [3] paved the way for RTs reduction to gain in the future. This scenario is supported by Reference [94] and by reversed application of Plotnikov’s [95] adaptive control without prior knowledge of parameters, or search of independent FHN model coefficients [96] and by estimating these very accurately for the models [97].

A second issue for further research is detailing the successive subprocesses with their neural fields. For instance, Chang and Jazayeri [98] introduced a multi-stimuli concept to explain time to contact a ball, that is the RT between seeing and catching a ball. Because the observed variability [99] of a 200 msec increase in RTs (depending on eye-blinks) has detailed processes accumulating sensorimotor RTs and practical application for research in driver behavior and sports. To gauge FHN models, Safaari et al. [100] modelled cortical state dynamics and single trial sensory processing.

The societal impact of this work shows that building robots will profit from sensorimotor modelling research, as exemplified by Ghadirzadeh et al.’s approach for self-learning of hand-eye coordination [101]. Virtual Reality is another application field of this research [102].

## Figures and Tables

**Figure 1 behavsci-09-00028-f001:**
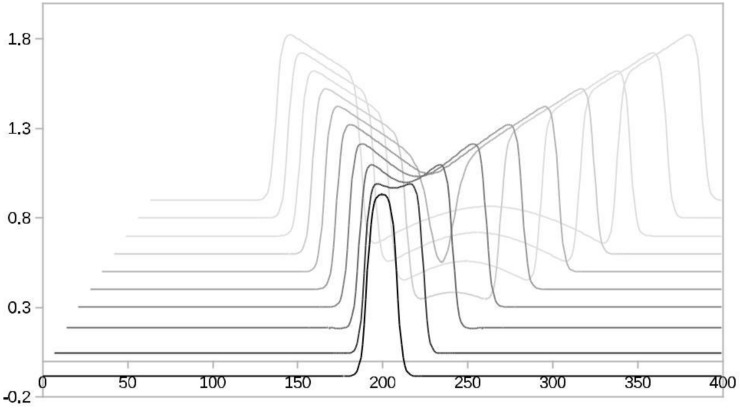
An FHN activator neuron’s initial spike splits and spreads in time. The time delays are visible by decreasingly paler black shades.

**Figure 2 behavsci-09-00028-f002:**
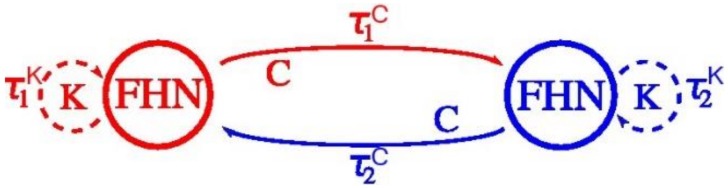
From Reference [45] and adapted from Figure 4 in Reference [20]. Two FitzHugh-Nagumo neuron groups with coupling strength *C* and internal feedback strength *K* exhibit different delay times forward and backward.

**Figure 3 behavsci-09-00028-f003:**
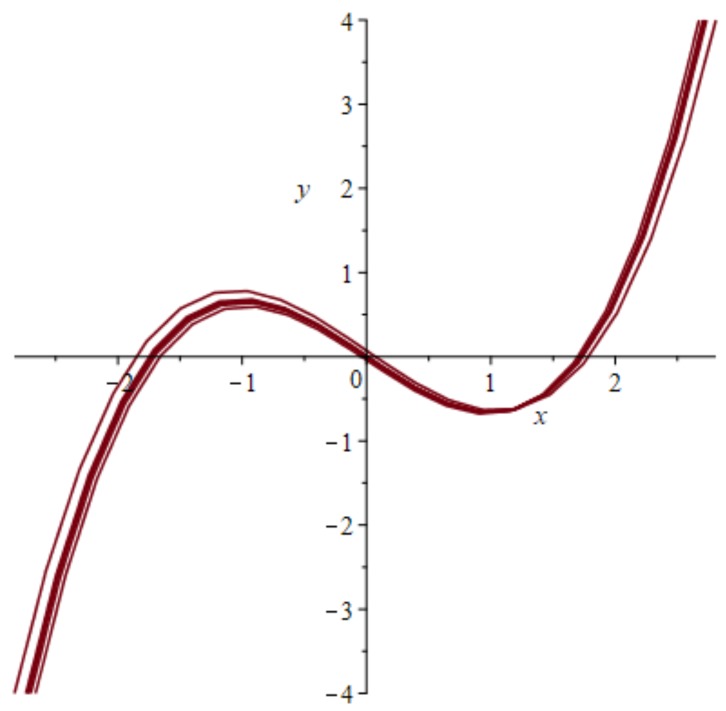
Scaled FHN Nullclines in the phase plane showing two stable points (zero-crossings of the third order Nullcline curve) and one unstable point in the origin.

**Figure 4 behavsci-09-00028-f004:**
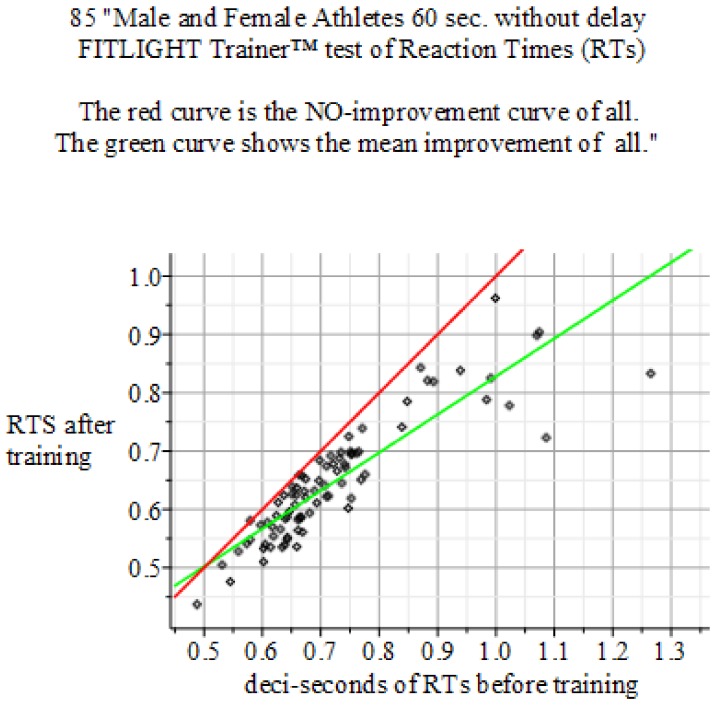
Summary of the above result: Reaction Times: 60 s without delay FITLIGHT Trainer™ equipment. Sample size of 85 men and women. Baseline sample mean 0.721 s, σ=±0.1075, and Final mean 0.646 s with σ=±0.0989. *P*-value 0.0000914155. Result: Hypothesis of NO-Effect of training is: Rejected.

**Figure 5 behavsci-09-00028-f005:**
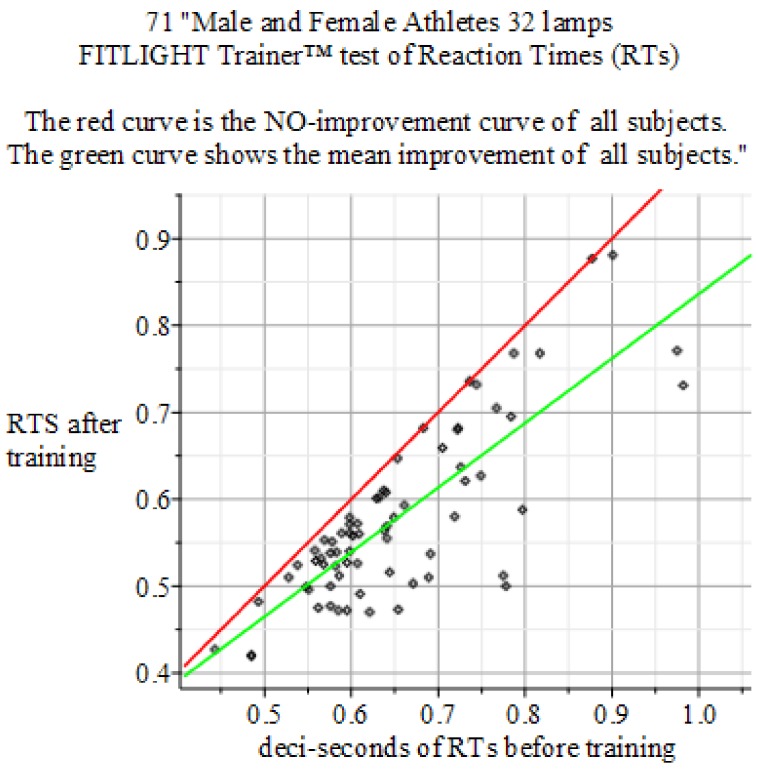
Summary of the above result. Reaction Times: 32 lamps FITLIGHT Trainer™ equipment. Sample size of 71 men and women. Baseline sample mean 0.650 s, σ=± 0.1075 and Final mean 0.577 s with σ=± 0.0988966. *P*-value 0.0000383454. Result: Hypothesis of NO-Effect of training is: Rejected.

**Figure 6 behavsci-09-00028-f006:**
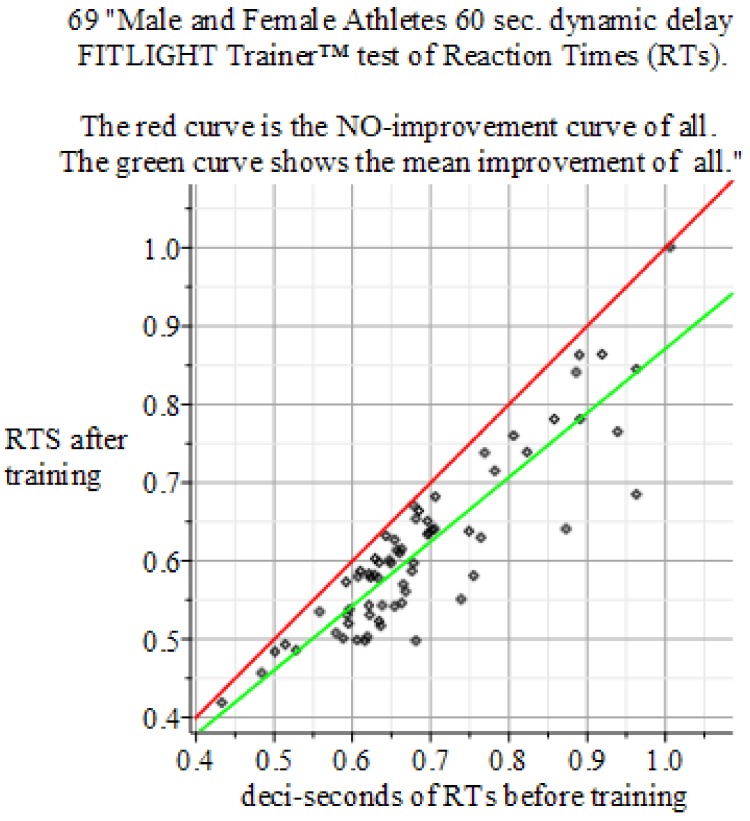
Summary of the above results. Reaction Times: 60 s dynamic delay FITLIGHT Trainer™ equipment. Sample size of 69 men and women. Baseline sample mean 0.687 s, σ=± 0.1194, and Final mean 0.614 s with σ=± 0.1099. *P*-value 0.0000914155. Result: Hypothesis of NO-Effect of training is: Rejected. The mean outcome of the 225 Reaction Time experiments is: Reaction Time performance improvement of +10.8% against the initial intake measurement.

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
