# Peer review of "Reaction Time Improvements by Neural Bistability"

_behavsci, 2019, doi:10.3390/bs9030028_

Reviewer 1 Report

This article reports interesting findings on the functional limits of response time regimes in eye-hand coordination supporting a neural delay integration model to account for the observed plasticity. The data are relevant and deserve publication. The introduction and presentation and discussion of the findings, however, lack in clarity. I recommend the authors throroughly revise these parts of their manuscript to ensure that the rationale is clear and can be understood also by non-experts in this field of specialty. The result section needs to be organized more clearly, with additional tables and clear explanations in legends and text, to be understood by scientists who are nnot familiar with the theoretical background.

Author Response

Dear Madam/Sir,

Upon the question "Does the introduction provide sufficient background and include all relevant references?"  your remark was: Must be improved! 
You also said: The introduction and presentation and discussion of the findings, however, lack in clarity. 
The improvement we did is to rewrite background text in the introduction and thereby elucidating this with accompanying references (about 30).

Are the results clearly presented? Not fully
We added an explanation in the text and for all: we added three plots to the results, to clarify the outcome of the significance tests of improved reaction times.

I recommend the authors thoroughly revise these parts of their manuscript to ensure that the rationale is clear and can be understood also by non-experts in this field of specialty. 
We followed this advice by working through the text and we did over 300 edits.

The result section needs to be organized more clearly, with additional tables and clear explanations in legends and text, to be understood by scientists who are not familiar with the theoretical background.
This provoking remark led us to three newly added plots in the results section. Specifically with concise but enlightening captures about the measurements.

Reviewer 2 Report

p.p1 {margin: 0.0px 0.0px 0.0px 0.0px; font: 12.0px 'Avenir Next'} p.p2 {margin: 0.0px 0.0px 0.0px 0.0px; font: 12.0px 'Avenir Next'; min-height: 16.0px}

Thank you to the authors - the paper was interesting, and quite though to get through due to some grammatical issues.

Below are some comments and suggestions:

Line 26

I would challenge the authors use of “proved”, as they themselves add the caveat that these are task dependent. A more nuanced “strong evidence for” would be more appropriate.

Line 31

The authors state the observations were done with high statistical significance. Statistical significance either is or isn’t as the significance point is set. I would recommend changing the sentence to something along the lines of:

Also (minor point) it is always better practice to state in long form what the abbreviation used stands for, e.g. MIEK. 

Alternative:

“A statistically significant improvement of eye-hand reaction times were observed following Mxx Ixx Exx Kxx (MIEK) training, as compared to (...)”

Line 39

A rewrite of this sentence would be encouraged as it exhibits grammatical issues.

Line 58

Grammatical issues in this sentence

Line 74

Grammatical issues in this sentence

Line 167

I would challenge the author on their use of the word bistable as I fail to see how it could both account for the sleep-wake cycle as well as bistable perception.

Line 125

I fail to see the logic of how changing behaviour leads to records being broken. A more nuanced statement would be recommended.

Author Response

Dear Madam/Sir,

Upon the question "Does the introduction provide sufficient background and include all relevant references?"  your remark was: Can be improved! 

The improvement we did is to rewrite background text in the introduction and thereby elucidating this with accompanying references (about 30).

Are the methods adequately described? Must be improved.

We added explanation in the text and we added three plots to the results, to clarify the empirical method.

Thank you to the authors - the paper was interesting,

Thank you!

The paper is quite though to get through due to some grammatical issues.

We followed this advice by working through the text and we did over 300 edits.

We resolved all the grammar issues with help of three native English speaking scholars (from Canada, USA and UK). 

Are the results clearly presented? Must be improved.

We included three extra graphical displays with the improved behavior of the athletes. To elucidate this further are the captures circumstantial.

Are the methods adequately described? Must be improved

We did by extensive rewrite and references.

Most important is your criticism: that you fail to see how the bi-stable mechanism of sleep-wake rhythms can be responsible for the reduction of reaction times. This was a very clear hint to us. This issue was important for us, and we cleared it up.
Profoundly.

You failed to see the logic of how changing behaviour leads to records being broken. A more nuanced statement would be recommended.

This provoking remark led us to extra illustrative explanation on the issue. We must confess that 'changing behaviour' is seen by us to include ' faster reacting'.

We thank you for your work! and help to improve our text.

Round  2

Reviewer 2 Report

The authors adequately addressed the points the I brought up.